# BepFAMN: A Method for Linear B-Cell Epitope Predictions Based on Fuzzy-ARTMAP Artificial Neural Network

**DOI:** 10.3390/s22114027

**Published:** 2022-05-26

**Authors:** Anthony F. La Marca, Robson da S. Lopes, Anna Diva P. Lotufo, Daniella C. Bartholomeu, Carlos R. Minussi

**Affiliations:** 1Electrical Engineering Department, UNESP—São Paulo State University, Av. Brasil 56, Ilha Solteira 15385-000, Brazil; anthony.marca@unesp.br (A.F.L.M.); anna.lotufo@unesp.br (A.D.P.L.); 2Computer Science Course, UFMT—Mato Grosso Federal University, Av. Valdon Varjão, 6390 Setor Industrial, Barra do Garças 78605-091, Brazil; robsonsilvalopes@gmail.com; 3Parasite Immunology and Genomics Laboratory, Institute of Biological Sciences, Minas Gerais Federal University, Belo Horizonte 31270-901, Brazil; daniella@icb.ufmg.br

**Keywords:** epitope mapping, diagnosis, in silico prediction, online training, hybrid approach

## Abstract

The public health system is extremely dependent on the use of vaccines to immunize the population from a series of infectious and dangerous diseases, preventing the system from collapsing and millions of people dying every year. However, to develop these vaccines and effectively monitor these diseases, it is necessary to use accurate diagnostic methods capable of identifying highly immunogenic regions within a given pathogenic protein. Existing experimental methods are expensive, time-consuming, and require arduous laboratory work, as they require the screening of a large number of potential candidate epitopes, making the methods extremely laborious, especially for application to larger microorganisms. In the last decades, researchers have developed in silico prediction methods, based on machine learning, to identify these markers, to drastically reduce the list of potential candidate epitopes for experimental tests, and, consequently, to reduce the laborious task associated with their mapping. Despite these efforts, the tools and methods still have low accuracy, slow diagnosis, and offline training. Thus, we develop a method to predict B-cell linear epitopes which are based on a Fuzzy-ARTMAP neural network architecture, called BepFAMN (B Epitope Prediction Fuzzy ARTMAP Artificial Neural Network). This was trained using a linear averaging scheme on 15 properties that include an amino acid ratio scale and a set of 14 physicochemical scales. The database used was obtained from the IEDB website, from which the amino acid sequences with the annotations of their positive and negative epitopes were taken. To train and validate the knowledge models, five-fold cross-validation and competition techniques were used. The BepiPred-2.0 database, an independent database, was used for the tests. In our experiment, the validation dataset reached sensitivity = 91.50%, specificity = 91.49%, accuracy = 91.49%, MCC = 0.83, and an area under the curve (AUC) ROC of approximately 0.9289. The result in the testing dataset achieves a significant improvement, with sensitivity = 81.87%, specificity = 74.75%, accuracy = 78.27%, MCC = 0.56, and AOC = 0.7831. These achieved values demonstrate that BepFAMN outperforms all other linear B-cell epitope prediction tools currently used. In addition, the architecture provides mechanisms for online training, which allow the user to find a new B-cell linear epitope, and to improve the model without need to re-train itself with the whole dataset. This fact contributes to a considerable reduction in the number of potential linear epitopes to be experimentally validated, reducing laboratory time and accelerating the development of diagnostic tests, vaccines, and immunotherapeutic approaches.

## 1. Introduction

To prevent the health system from collapsing and millions of people dying each year, it is important to develop accurate diagnostic tests that help with control and monitoring, as well as efficient vaccines that immunize people against dangerous and highly infectious diseases.

Vaccination has been applied systematically for decades, culminating in the eradication of many infectious diseases, such as mumps, rubella, whooping cough, polio, measles, and tetanus [1,2]. In addition, vaccination is also an efficient approach to avoiding or reducing antibiotic prescriptions and, consequently, mitigating the emergence of increasingly resistant bacterial strains.

To develop these vaccines and offer greater safety, potency, and efficacy [3], it is first necessary to identify the highly immunogenic regions within a given protein of pathogenic organisms. These regions that represent the interface between the pathogen and the immune response are known as epitopes of B- and T-cells, and are responsible for inducing an immune response [4].

B-cell epitopes are classified as linear or conformational. Linear epitopes are formed by sequential (contiguous) amino acids of the protein, while conformational epitopes are composed of amino acids that are not contiguous in the primary sequence but are joined in the folding of the protein [5].

Although most epitopes are conformational, their identification requires information based on their three-dimensional structure, which is still very limited compared to the number of protein sequences currently available in public databases [4]. Therefore, this study focuses on the prediction of linear epitopes of B-cells.

The conventional and most reliable methods used for mapping these epitopes, such as crystallography [6] and nuclear magnetic resonance techniques [7], are expensive and require extensive laboratory work. Therefore, silicic identification methods have been widely used for the development of computational models that can more effectively predict the presence and location of linear epitopes of B-cells, from an amino acid sequence of a pathogenic organism [8].

Due to the simplicity in generating embeddings, the study of prediction of linear epitopes of B-cells has been taking place for several years, and already has many tools available in the literature, in which they use machine learning techniques in their predictions, such as ABCPred [9], BCPred [10], SVMTrip [11], BepiPred-2.0 [12], and EpiDope [13].

However, because epitopes do not have intrinsic characteristics in the protein, but only when they interact with the antigen paratope, the task of predicting epitopes in silico becomes challenging [5]. This fact reflects negatively on the performance of current tools, limiting them in the accurate identification of linear epitopes of B-cells.

Another factor that influences performance is the low quantity and quality of data available for training. Thus, with the constant updating of one of the main public databases, the Immune Epitope Database (IEDB) [14], together with new trends in feature extraction and new artificial intelligence techniques, it is expected that the accuracy of the identification of linear epitopes of B-cells, and of new tools, will gradually reach levels of excellence.

Two tools frequently used to predict linear epitopes of B-cells are BepiPred-2.0 [12] and EpiDope [13]. According to the authors, the first tool reached an AUC of the ROC curve of 0.57, while the second reached 0.605. These values are on the same level as the predictions provided by the IEDB (http://tools.iedb.org/main/bcell, accessed on 2 March 2022), demonstrating the difficulty that these tools have in their predictions.

In this context, we developed BepFAMN, a tool that uses a new approach to identify and locate linear epitopes of B-cells in amino acid sequences. The tool uses an ANN from the ART (adaptive resonance theory) family that allows online training and promotes great ease in learning new patterns, without losing the memory of already learned patterns (plasticity and stability).

The ART family networks are very suitable for complex classification problems, where useful knowledge about the features of the objects to be classified is quite limited [15]. With enough data, a Fuzzy-ARTMAP ANN can automatically recognize the appropriate features for classification, making it suitable for predicting linear B-cell epitopes.

In the tests of the tool, the AUC of the ROC curve and the AUC of the precision-recall curve were used, because, according to [16], in situations where the classes are unbalanced (the amount of negative epitopes is much higher than the number of positive epitopes, for this study), the second measure provides a more realistic view of the algorithm’s performance.

For the target problem, the number of false positives and false negatives impact our model in different ways. In this way, we use the F1-score metric, in the test dataset, to create a result from these divergences.

BepFAMN was able to identify linear B-cell epitopes with a ROC curve AUC of 0.7831 and a precision-recall curve AUC of 0.8343, which significantly exceeds previous methods (EpiDope: 0.605 and 0.660, respectively). This helps to considerably reduce the number of potential linear epitopes to be experimentally validated, reducing costs and services, as well as accelerating the process of vaccine development and diagnostic tests.

## 2. Related Work

The ABCPred [9] used recurrent neural networks (RNN), with 16 neurons in the input layer and 35 neurons in a single hidden layer, to predict linear B-cell epitopes. The dataset was obtained from Bcipep [17], and contained 2479 positive epitopes of variable size. To standardize the size of the epitopes and facilitate RNN learning, all redundant epitopes formed by more than 20 amino acids were removed, resulting in a final set of 700 positive epitopes. Negative epitopes, with a fixed length of 20, were randomly extracted from Swiss-Prot [18]. Sliding windows ranging from 10 to 20 amino acids were used around each residue to obtain its properties and, thus, were presented to the RNN. For the tests, the 5-fold cross-validation was used, obtaining an accuracy of 66% and MCC of 0.3128, with a sliding window of size 16 [9].

The BCPred [10] used SVM with five Kernel variations and fivefold cross-validation to predict linear B-cell epitopes. It used a reduced dataset containing 701 positive epitope samples, extracted from Bcipep [17], and 701 negative epitopes, randomly extracted from Swiss-Prot [18]. At the end of the tests, using the radial-based kernel for the SVM and a sliding window of size 20 to obtain the amino acid properties, the proposed method generated promising results, reaching an AUC of approximately 0.76 [10].

The BayesB [19] is a proposed method to predict linear B-cell epitopes based on protein structure. The method uses SVM and Bayes feature extraction to predict epitopes of different sizes (12 to 20). Two sets of reference data [10,20] were used. The first base contained 701 epitopes with reduced length homology to 5 from different peptides (12, 14, 16, 18, and 20). The second contained 872 1-size-fits-all epitopes (20). In both sets, equal amounts of negative epitopes were randomly removed by the authors from the Uniprot database [21]. The feature vectors were coded in a bi-profile manner [22], containing positive-position-specific profiles and negative-position-specific profile attributes. These profiles were generated by calculating the frequency of occurrence of each amino acid at each position of the peptide sequence in the set of positive and negative epitopes. In this way, each input peptide (size 20, for example) would be encoded by a vector of dimension 40 (20 × 2), which contains information about the residues in the positive spaces (epitopes) and in the negative spaces (non-epitope). For the tests, 10-fold cross-validation was used, obtaining a precision of 74.5% [19].

The SVMTriP [11] is a method of predicting linear antigenic B-cell epitopes, which combines SVM with tri-peptide similarity and propensity scores. The dataset was extracted from the IEDB [23] and contained 65,456 positive epitopes. After grouping them according to the degree of similarity (greater than 30%) measured by BLAST, and eliminating the redundant epitopes, the resulting dataset consisted of 4925 positive epitopes. Negatives were extracted from non-epitope segments of the corresponding antigen sequences, keeping the same number of subsequences in length. The SVMTriP achieved a sensitivity of 80.1%, a precision of 55.2%, and an AUC value of 0.702, using 5-fold cross-validation [11].

The BEEPro [24] is a proposed method to predict linear and conformational B-cell epitopes through evolutionary information and propensity scales. This method uses 16 properties for the construction of the feature vector, which includes the position specific score Matrix (PSSM), an amino acid ratio scale, and a set of 14 physicochemical properties, obtained through a process of feature selection. For training, SVM with the radial base kernel, five-fold cross-validation, and the following seven datasets were used: Sollner [25], AntiJen1 and AntiJen2 [26], HIV [27], Pellequer [28], PC [29], and Benchmark [30]. These were used to avoid bias and distortions in the results. The BEEPro achieved an AUC and accuracy ranging from 0.9874 to 0.9950 and 93.73% to 97.31%, respectively [24].

The SVM BEEProPipe [31] is a linear B-cell epitope prediction tool, which uses a different rate for training and validation (bacteria, viruses, and protozoan). The dataset used was extracted from IEDB [23], using positive, negative, and non-epitopes. Positives and negatives are epitopes and sequences that proved not to be an epitope, respectively, after experimental validation. Non-epitopes are amino acid sequences that have not been tested experimentally, but are derived from the same proteins that present positive epitopes. To form the final datasets, redundant epitopes were eliminated using the BLAST tool (80% similarity). In the end, a total of 5548 positive epitopes and 5882 negative epitopes were selected, creating 2 databases, 1 with positive and negative epitopes, and the other with positive and non-epitopes, following a balanced distribution in each class (50%). As a prediction strategy, this tool used 14 physicochemical properties, the proportion of amino acids in each protein, and the radial kernel SVM. To create the feature vector (embeddings), a 20-size sliding window was used, centered on each amino acid. Then, 5-fold cross-validation was used for the tests, obtaining a precision of 95.71%, 94.90%, 92.01%, 94.94%, 89.09%, and 88.74% for the datasets of protozoa, bacteria, and virus, respectively, and for positive/negative epitopes and positive/non-epitope epitopes, respectively [31].

The BepiPred-2.0 [12] is a web server used to predict B-cell epitopes from antigen sequences. The database used was obtained from the IEDB [23] and contained 11,834 positive epitopes and 18,722 negative epitopes. Peptides smaller than 5 or greater than 25 amino acids have been removed, as epitopes are rarely outside this range. The method used for training was the “Random Forest Regression” (RF) algorithm with fivefold cross-validation. Each residue was coded using its volume, hydrophobicity, polarity, relative surface accessibility, and secondary structure, in addition to the total volume of antigen, generated by the individual sum of each residue. In the pre-processing step, a nine-size sliding window was used, centered on each residue, throughout the entire peptide. According to the authors, BepiPred-2.0 surpassed (on average) all the main tools for predicting linear B-cell epitopes, hitherto available in the literature [10].

EpiDope [13] is a tool developed in Python that uses deep neural networks (DNN) to detect regions of B-cell epitopes in individual protein sequences. The dataset was extracted from the IEDB [23] and originally contained 30,556 protein sequences, each of which contained experimentally verified epitopes or negative epitopes. To better represent the dataset, identical protein sequences were merged, but information about their verified regions was kept. The EpiDope architecture consists of two parts. The first uses context-sensitive amino acid embeddings, each of which are encoded by a 1024 length vector that encodes physicochemical and structural information. These embeddings are previously calculated by ELMo [32] and are the input to a bidirectional LSTM (long short-term memory) layer (2 × 5 nodes), followed by a dense layer with 10 nodes. The second encodes each amino acid in a context-insensitive vector of length 10, which is connected to a bidirectional LSTM [32] layer (2 × 10 nodes), followed by a dense layer with 10 nodes. At the end of the structure, both layers are connected to an additional dense layer containing 10 nodes, which connects to an output layer with 2 nodes, 1 representing the positive epitopes and the other the negative epitopes. For validation, 10-fold cross-validation was used, obtaining an AUC of 0.605, against 0.465, obtained by the BepiPred-2.0 tool, which, according to the author, was the current leader in the literature [13].

## 3. Materials and Methods

In this section, the dataset and the methodology used for training the proposed tool are presented and briefly described, in addition to the validations performed.

### 3.1. Training and Testing Dataset

The first stage of this research consisted of searching for linear B-cell epitopes, available in public databases. Due to the quantity, diversity, experimental validation, and many fine filters of research available in the IEDB database, it was decided to provide both positive and negative epitopes to the training and testing experiments.

Positive epitopes are those that have been experimentally validated and have positive assays (results) in at least one experiment. The negative ones were also experimentally validated and must not present a positive test in any experiment.

Obtaining the positive and negative epitopes of the IEDB was carried out on 5 April 2021, using the following filters for the query: linear epitopes, infectious diseases of humans, B-cell assays, no MHC restriction, and bacteria, viruses, and protozoa.

In the end, the database contained 11,509 positive epitopes and 28,080 negative epitopes, as described in Table 1.

Four databases are created, consisting of three independents, each formed by the residues of a taxon, and one dependent, formed by the union of the residues of all three taxa. The amount of positive and negative epitopes of each base is shown in Table 1 and are named as follows:DB_bac: composed only of positive and negative bacterial epitope residues;DB_vir: composed only of positive and negative virus epitope residues;DB_prot: composed only of positive and negative epitope residues from protozoa;DB_all: composed of positive and negative epitope residues from all three taxa.

A very important piece of data that was missing in the IEDB database metadata is the complete protein sequence of each antigen. Because machine learning algorithms use the context of a protein’s amino acids to try to find correlations and predict possible epitopes, it was necessary to obtain this information. With the code of each antigen, obtained from the IEDB database, the NCBI database [33] was accessed to retrieve this information.

### 3.2. Validation Dataset

The validation database is independent of the training/testing database and was taken from [12]. The base contains 30,556 protein sequences, where each sequence contains a tagged region, which represents a positive epitope or a negative epitope, both experimentally validated.

Among the 30,556 sequences, 11,834 are positive epitopes and 18,722 are negative epitopes. The subset of positive epitopes has an average length of 13.99 (number of consecutive amino acids), while the subset of negative epitopes has an average length of 13.20, as shown in Table 2 [13].

### 3.3. Preparation of Training and Testing Data

The epitopes that had similarities greater than or equal to 80% [31], evaluated by the BLAST software [34], were grouped. Only one of each group was randomly selected and kept as an epitope sequence in the final dataset. This procedure is necessary to prevent the machine learning algorithm from memorizing very similar epitope sequences and favoring generalization.

Next, it was necessary to perform some adjustments to eliminate irrelevant data, in addition to enriching and transforming essential data into “cleaner” and “friendly” data. Thus, in addition to epitope information, only the metadata about the initial and final position of each epitope and the antigen sequence was kept. To this dataset, only one new metadata was added, the complete protein sequence of the antigen.

Initially, a great imbalance between the classes was found. To prevent the knowledge models generated from being biased and important epitopes from being discarded, prevalence correction techniques were carefully applied, such as Stratified Sampling, for example. Furthermore, it was identified that some records had missing important data and, therefore, were also eliminated. Other records removed were those with epitopes greater than 30 and smaller than 5 amino acids, as they occurred sporadically in the database. In the end, the total database was composed of 9,968 positive epitopes and 10,766 negative epitopes, as described, by taxon, in Table 3.

### 3.4. Preparation of Validation Data

From the raw dataset of [12], a set independent from the one used in the training and testing steps, all the pre-processing steps of [13] were applied to generate a database identical to the one used, in order to compare the results between the studies. Figure 1 presents the entire pre-processing process performed on the referred dataset [13].

Firstly, the amino acid sequences of identical proteins were joined together, but keeping the information about their verified regions (30,556) (Figure 1A,B), resulting in a reduced dataset of 3158 proteins. Then, to attenuate the redundancy of very similar amino acid sequences, clusters of all sequences were generated with the CD-HIT tool [35], using a threshold of 0.8 (Figure 1C). As a result, 1798 groups of sequences were generated, in which, from each group, only the sequence that contained the highest number of verified regions was retrieved (Figure 1D), resulting in 24,610 regions. The grouping step was again applied to the reduced dataset, but used a threshold of 0.5 (Figure 1E), resulting in 1378 groups of sequences [13].

It is important to mention that this entire procedure is necessary to avoid overrepresentation in the data, which can directly influence the training of the machine learning algorithm.

### 3.5. Prediction Strategy

The prediction strategy of this work was inspired by the BEEPro tool [24], which presented promising results. The purpose of the tool is to predict linear and conformational B-cell epitopes using SVM in 16 properties, namely specific position scoring matrices (PSSM), amino acid proportion scale, and a set of 14 physicochemical properties obtained from the AAindex database [24].

As PSSM calculations for very large proteomes are time-consuming, with any possibility of providing a web-based epitope prediction tool, this work chose not to use PSSM, and only use the other properties.

Several studies in the literature [11,19] showed that certain amino acids (di- or tri-peptides) occur with greater or lesser frequency in epitopes. According to Wee et al. [19], the amino acids tryptophan, proline, and glutamine are found more frequently in positive epitopes, whereas phenylalanine and leucine are found less frequently. As for Yao et al. [11], amino acids, such as glutamine and proline, play a fundamental role in the identification of epitopes, as they appear more frequently in tri-peptides.

Therefore, the proportion rate was calculated for each of the 20 amino acids present in the database, through Equation (1). However, to determine how often each amino acid occurs in positive and negative epitopes, Equation (1) [24] was applied individually, first to positive epitopes and then to negative ones.
(1)$$p_{\alpha i}=\frac{f_{\alpha i}^{+}/\sum_{i}f_{\alpha i}^{+}}{f_{\alpha i}^{-}/\sum_{i}f_{\alpha i}^{-}},$$
where:
$f_{\alpha i}^{+}$: frequency of the *α*_*i* amino acid in positive epitopes of the protein;$f_{\alpha i}^{-}$: frequency of the *α*_*i* amino acid in negative epitopes of the protein.

To attenuate the frequency dominance of some amino acids and to prevent the machine learning algorithms from becoming biased, Equation (2) [31] normalizes them in an interval between [0, 1].
(2)$$p_{\alpha i}=\left[\;\frac{p_{\alpha i}-\min\left(p_{\alpha i}\right)}{\max\left(p_{\alpha i}\right)-\min\left(p_{\alpha i}\right)}\;\right],$$
where:
$\max\left(p_{\alpha i}\right)$: highest frequency value among amino acids;$\min\left(p_{\alpha i}\right)$: lowest frequency value among amino acids.

After testing a combination of various physicochemical and biochemical properties in predicting linear B-cell epitopes, reference [24] found that 14 properties stood out over the others. They are as follows: hydrophilicity (PARJ860101), hydrophobicity (PONP930101), flexibility (KARP850102 and BHAR880101), interactivity (BASU050101), composition (GRAR740101), volume (GRAR740103), charge transfer (CHAM830107), ability to donate charge transfer (CHAM830108), ability to donate hydrogen bonds (FAUJ880109), frequency of the alpha-helix structure (NAGK730101), beta structure frequency (NAGK730102), coil structure frequency (NAGK730103), and antigenicity [36]. This fact justified the use of these same properties in the current study.

Thus, having obtained the values of these properties for each amino acid and their respective ratio of positive and negative epitopes, it is necessary to scan the epitopes of interest, within the complete antigen sequence, to generate the attributes that will be presented to the machine learning algorithm. For this purpose, the sliding window method is used, which uses a moving window, assigning an average to the central amino acid of the window, in property *j*, for *j* = 0, 1, ..., 14, according to Equation (3) [24].
(3)$${AvgScale}_{j}=\frac{\sum_{i}\left(1\;–\;f\;*\;\left|c\;–\;i\right|\right)\;*\;S_{i}}{w},$$
where:
*i*: index of the position of the residue in the sliding window;*c*: index of the position of the central residue of the window;|c – i|: distance in number of residues between residue *©* and the central residue *c*;*f*: linear weight factor (assigned value);$S_{i}$: physicochemical property value or amino acid ratio of the residue at position *i*.

To define the size of the sliding window, the mean length of positive and negative epitopes of each taxon and the general mean were observed, as shown in Table 4. Table 4 also shows the respective variance.

The tests were carried out empirically, using sizes 10, 12, 15, 17, and 20 for the sliding window. In the end, the one that presented the best results was the sliding window with a size equal to 20, and this was, therefore, the one used in the work.

The value assigned to the linear weight factor (f) was the value of 0.08 recommended by the study [24]. The purpose of this factor is to increase the importance of the neighboring amino acids, in relation to the analyzed amino acid (the central one), as they are close together.

After applying all these operations mentioned above, the following data format results in Equation (4):<*class*, *value*_0_, *value*_1_, …, *value*_*n*−1_, *value*_*n*_>,(4)
where:< >: an epoch used in the training process;Class: an integer indicating the class the instance belongs to, 1 for positive epitopes and 0 for negative epitopes;*n*: number of properties;*value_i_*: a real value, which represents a physicochemical property or the ratio of the amino acid *i*.

To prevent attributes with very large numerical values from preponderating very small values, all attribute values have been normalized between [0, 1].

It is important to point out that each instance represents the calculations of the 15 properties applied to an amino acid of interest, being either for the subset of positive epitopes or for the subset of negative epitopes.

After all the pre-processing procedures were applied to the datasets, the cross-validation technique was used, with K-folds equal to 5, to divide each dataset into k mutually exclusive subsets of the same size and, from then, to use a subset k for testing and the other subsets (k−1) for training.

### 3.6. Prediction Strategy

The FAM ANN technique has been used quite frequently in prediction work, mainly in the field of electrical engineering, because of its plasticity, stability, and ability to converge a few times [37,38,39].

Although FAM ANN is a technique that presents good generalization rates, the performance of its prediction depends on the choices of the values of the following vigilance parameters: *ρ_a_*, *ρ_b_*, and *ρ_ab_*. An inadequate choice can result in a loss of accuracy of the results, as values close to zero allow little identical patterns to be grouped into the same category, and values close to one allow small variations in input patterns, leading the ANN to create new classes [15].

In this research, values of 0.61 (*ρ*_abaseline_) 0.8, and 0.99 were used for the parameters *ρ_a_*, *ρ_b_*, and *ρ_ab_*, respectively.

The values for the α and β parameters are set equal to 0.1 and 1.0, respectively. The rate of increment of *ρ_ab_* was defined through empirical tests and fixed equal to 0.1.

The weight matrices *w_a_*, *w_b_*, and *w_ab_* were started with a value equal to 1, indicating that all activities were inactive. Initially, the matrices have only one row, indicating that at the start of learning there is only one active neuron in each matrix. As the training process takes place and activities begin to activate, new neurons are dynamically created and initialized. It is worth mentioning that each weight matrix must follow the dimensions defined by Equation (5). Furthermore, the number of neurons must be less than or equal to the values of the parameters *m_a_*, *m_b_*, and *n*.
(5)$$w_{a}=\left(n\times m_{a}\right),w_{b}=\left(n\times m_{b}\right),w_{ab}=\left(n\times n\right),$$
where:
$m_{a}$: number of components (attributes) of the input vectors;$m_{b}:$ number of output vector components;*n*: number of input patterns.

It is important to point out that the complements of the input and output patterns are also accounted for when defining the values of the *m_a_* and *m_b_* parameters, and, therefore, the values of these parameters must be doubled.

In this work, as already mentioned, 15 properties were used to predict whether an amino acid of an antigen protein is an epitope (1) or if it is not an epitope (0). In this way, the values of the parameters *m*_a_ and *m_b_* were defined in 30 and 2, respectively. The value of n was set to the values of 35,157, 97,001, 109,120, and 241,278 for the DB_bac, DB_prot, DB_vir, and DB_all datasets, respectively.

As it is a binary classification problem, the *w_b_* weight matrix was optimized and defined according to Equation (6).
(6)$$w_{b}=\left(2\times m_{b}\right),$$

#### 3.6.1. Training

The order of presentation of the ANN input and output patterns can be used sequentially or in a random/pseudorandom order. However, from a mental point of view, it is more sensible to adopt the random/pseudorandom order. Therefore, in this work, the second approach was adopted.

To calculate the complement of the matrix pairs (input and output) the strategy of doubling the number of rows of the respective matrices instead of doubling the number of columns was used. Thus, if line zero of the ***w****_a_* weight matrix contains the first input pattern, line one contains its complement. This process is repeated for all other input and output patterns, and at the end, the even lines will contain the input or output patterns, and the odd lines their respective complements.

Using the training parameter *β* = 1 (fast training) each knowledge model was generated from a single epoch. To improve the prediction of epitopes, for each fold created by cross-validation, three knowledge models are generated, and for the training of each one, the input and output patterns are presented in a pseudo-random way and in a different order, allowing that each model learns in a way and generates different amounts of neurons.

Table 5 presents the number of neurons that were created for each knowledge model for each partition of each dataset, after the training and cross-validation process. It is noteworthy that this quantity is only for the *w_a_* and *w**_ab_* weight matrices, since the *w_b_* weight matrix has only two neurons (a binary classification problem, as discussed above).

To generate each group of knowledge models for each partition of each dataset, the same quantities and the same input data were used, changing only the order in which the inputs were presented to the ANN.

#### 3.6.2. Diagnosis

In the diagnosis, ANN activates the category that best represents the entry pattern, through the choice function, and checks whether the degree of similarity between the entry pattern and the active category meets the value defined in the parameter *ρ_a_*. However, it is possible that input patterns appear with very different characteristics from the existing categories, so none of the categories are similar enough to meet the vigilance test.

For these cases, the strategy of automatic gradual decrement of parameter *ρ_a_* by parameter ε was adopted, until the closest category became compatible enough and passes the vigilance test. Empirically, the value of ε was set at 0.1.

After passing the vigilance test, the value of the *ρ_a_* parameter returns to its original value of 0.61 and the execution passes to the inter-ART module. At that moment, it is verified which column of the *w_ab_* weight matrix is active, based on the index of the winning neuron of the *w_a_* weight matrix (in this same row in *w_ab_*). With the *w_ab_* column index, it is verified which category (row) belongs to this index in the *w_b_* weight matrix, this category being the final diagnosis of the FAM ANN.

Figure 2 shows, in a simplified way, the implemented FAM ANN diagnosis process. In the diagnosis, ANN activates the category that represents the best entry pattern, through the choice function, and checks whether the degree of similarity between the entry pattern and the active category meets the value defined in the parameter *ρ_a_*. However, it is possible that input patterns appear with very different characteristics from the existing categories, so none of the categories are similar enough to meet the vigilance test.

For these cases, the strategy of automatic gradual decrement of parameter *ρ_a_* by parameter ε was adopted, until the closest category became compatible enough and passes the vigilance test. Empirically, the value of ε was set at 0.1.

For the diagnosis process, cross-validation and the competitive strategy were used, in which each generated knowledge model (three) performs its own process, and in the end, there is competition with its results, such as the result that prevails, is the final diagnosis of ANN, on the current input pattern. It is noteworthy that this approach was applied to each dataset independently, and only the input patterns were presented to the ANN, meaning that the output patterns were used only to validate the prediction results. 

For the validation database, the same competition process was applied, but in 100% of the dataset, in addition to using the same knowledge models generated by the training dataset DB_all.

### 3.7. Evaluation Measures

To measure the performance of the knowledge models generated for each dataset, the following metrics commonly used to assess the quality of prediction tools are used: sensitivity, specificity, precision, accuracy, and MCC, which are defined, respectively, by the Equations (7)–(11) [31,40].
(7)$$R_{\mathrm{sen}}=\frac{\mathrm{TP}}{\left(\mathrm{TP}\;+\;\mathrm{FN}\right)},$$
(8)$$R_{\mathrm{esp}}=\frac{\mathrm{TN}}{\left(\mathrm{TN}\;+\;\mathrm{TF}\right)},$$
(9)$$R_{\mathrm{pres}}=\frac{\mathrm{TP}}{\left(\mathrm{TP}\;+\;\mathrm{FP}\right)},$$
(10)$$R_{\mathrm{acc}}=\frac{\left(\mathrm{TP}+\mathrm{TN}\right)}{\left(\mathrm{TP}\;+\mathrm{FP}\;+\;\mathrm{TN}\;+\;\mathrm{FN}\right)},$$
(11)$$R_{\mathrm{mcc}}=\frac{\left(\mathrm{TP}\;\times \;\mathrm{TN}\;-\;\mathrm{FP}\;-\;\mathrm{FN}\right)}{\sqrt{\left(\mathrm{TP}\;+\;\mathrm{FP}\right)\;\left(\mathrm{TP}\;+\;\mathrm{FN}\right)\;\left(\mathrm{TN}\;+\;\mathrm{FP}\right)\;\left(\mathrm{TN}\;+\;\mathrm{FN}\right)}},$$
where:TP: number of true positives. Residues predicted as epitopes and actually are epitopes;TN: number of true negatives. Residues predicted as negative epitopes and actually are negative epitopes;FP: number of false positives. Residues predicted as positive epitopes and are negative epitopes;FN: number of false negatives. Residues predicted as negative epitopes and are positive epitopes;R_sen_: the rate of positive epitopes that are correctly predicted as positive epitopes;R_esp_: the rate of negative epitopes that are correctly predicted as negative epitopes;R_pres_: among all epitope classifications, how many are correct. Ideal in situations where FPs are considered more harmful than NFs;R_acc_: indicates overall performance. How well did the model rank;R_mcc_: is a measure considered balanced and was proposed by Matthews in 1975. It assumes values between −1 and 1, where:
Value close to −1, corresponds to a bad prediction;Value close to 0, corresponds to a random prediction;Value close to +1 corresponds to an excellent prediction.

In addition to the aforementioned metrics, a graphical method is used, capable of demonstrating the relationship with sensitivity and specificity, called ROC curves (receiver operating characteristics). However, to make it possible to compare the results of the classifiers, it is necessary to reduce the ROC curve to a simple scalar. The method usually used for this purpose is called AUC, the function of which is to calculate the area under the ROC curve.

### 3.8. Assessing Approaches to Predicting Epitopes in the Validation Dataset

For the validation dataset, the performance of the proposed tool is compared with the EpiDope tool [13] and with three more frequently used tools in the prediction of linear B-cell epitopes, in their most recent versions and available on the IEDB. These are BepiPred-2.0 [12], and Chou and Fasman beta-turn prediction [19].

Although all the metrics mentioned in Section 3.7 are calculated, when comparing the tools, only the ROC and precision-recall curves are used. According to [16], ROC curves can present an overly optimistic view of the performance of an algorithm, in cases where there is a distortion in the distribution of classes. A widely used and efficient alternative in these situations is the precision-recall curve. In this study, ROC curves and precision-recall curves are calculated using the scikit-learn library [41].

## 4. Results

### 4.1. Test Dataset

After the process of diagnosis and possession of the results from the DB_bac, DB_vir, DB_prot, and DB_all databases, the metrics discussed in Section 3.7, were applied to each result. Figure 3 presents an overview of the metric/dataset relationship.

According to Figure 3, the results of all metrics were above 83%, except for the MCC value for the DB_vir database, which was 70.94%. Table 6 presents, in detail, the result of each metric on each dataset.

Seeking to assess the ability of the implemented method to identify positive epitopes, an average of the ROC curve is calculated for each subset of each database. This was necessary, as five-fold cross-validation is used and, therefore, for each partition, a ROC curve is calculated, and at the end, an average of the ROC curve is generated. Figure 4 shows the average for the ROC curves of the DB_bac (a), DB_vir (b), DB_prot (c), and DB_all (d) databases.

All sets used presented a ROC curve with accuracy greater than 86%, and the protozoan database (DB_prot) presented the best accuracy index, at approximately 99.45%. The worst rate was revealed by the viral database, a figure of approximately 86.89%. The other sets were within this range, obtaining approximately the values of 94.54% and 92.89% for DB_bac and DB_all, respectively.

### 4.2. Validation Dataset

The validation database was used solely and exclusively to validate the knowledge models generated by the DB_all dataset. This way, there is no need to combine several forecasts to generate an average of the ROC curve, since the cross-validation technique is not applied and the data is not used for training. At this stage, the 15 knowledge models were used, together with the competition technique, to predict the linear B-cells epitopes.

From the ANN diagnosis, all the metrics briefly discussed in Section 3.7 are applied. Figure 5 presents, in detail, the values achieved.

According to Figure 5, it can be seen that the model obtained promising indexes for all the metrics used [3,12,13]. The MCC with the value of 0.5674 indicates an average prediction, superior to a random one. The other metrics obtained very close values, between 74 to 82%. The ROC curve (a) and the precision-recall curve (b) reached values of approximately 0.7831 and 0.8343, respectively, as shown in Figure 6.

The F1-score represents a harmonic mean between precision and recall, combining both in order to bring a single value that indicates the overall quality of the model [42]. To present this general perspective, this metric was calculated for the dataset, and reached a value of approximately 0.7884, indicating the balance of the model in the correct prediction of positive epitopes and non-epitopes. In this study, F1-score was calculated using the scikit-learn library [41].

### 4.3. Comparison of Results

The main tools available in the literature do not allow for large-scale testing and limit access to many features. Thus, the results of reference [13] are used for comparison purposes.

In this work, the AUC of the ROC curve and the precision-recall curve are used to compare the performance of EpiDope with the performance of other tools commonly used in predicting linear B-cell epitopes.

From the results obtained with the validation database, a set of data that went through all the pre-processing procedures, as described in reference [13], was tried to compare with the results of the other tools.

Table 7 presents the AUC results of the ROC curve and the precision-recall curve of the developed tool, the aforementioned tools, and a random mean prediction. Note that the proposed tool reached a value of approximately 0.7831 for the AUC of the ROC curve and 0.8343 for the AUC of the precision-recall curve, surpassing the values of the other tools (about 18% of the best result), including a random mean prediction.

## 5. Discussion

There are several computational tools available in the literature for in silico identification of linear epitopes of B-cells. As described in Section 3, machine learning techniques, such as SVM, ANN, and hidden Markov models, combined with another approach, usually with the propensity scale method [24], are commonly used for this purpose.

However, many of these tools have a low rate of accuracy in predicting epitopes, or when they improve the results, they involve high computational costs, making large-scale use unfeasible, especially via the Internet.

One factor that directly influences the reduction of these results is the bias inherent to this type of dataset. To obtain an unbiased epitope prediction it is essential to have a wide variety of known epitopes and non-epitopes from different organisms in the training set. Therefore, analyzing the taxonomic variety of the proteins provided by the database (in this work, the IEDB), eliminating redundancy, and balancing the classes is fundamental.

In the extraction of training data, a similarity reduction filter (BLAST) was used, with a threshold of 80%, along with prevalence correction techniques to attenuate the number of redundant epitopes and balance the amount of data in each class to avoid bias to ANN. The final training dataset contained 18.81% bacteria, 42.69% viruses, and 38.50% protozoa. Overall, the training data exhibited a sufficient degree of taxonomic diversity.

The same process occurred for the test data. However, these data were taken from the works [12,13]. To reduce similarity, the CD-HIT tool was used, with 2 thresholds, 0.8 and 0.5, applied sequentially, respectively. The final validation dataset had 20.6% bacteria, 25.1% viruses and 54.2% protozoa. It is worth mentioning that the data from this set are formed by taxa from any host, available in the IEDB, that generates an immune response. On the other hand, the training dataset is formed only by the “Human” host taxons.

The method implemented for epitope prediction, as mentioned in Section 3, was inspired by the works [24,31]. We chose not to use evolutionary information, coded as PSSM [24], due to its high computational cost, which could make it difficult to access the tool remotely. However, even without using it, the Fuzzy-ARTMAP ANN showed promising results, both for the training dataset and for the validation set.

The Fuzzy-ARTMAP ANN was trained and tested on datasets formed by a single taxon and one formed by the union of the three taxa. For each set, 5 times cross-validation and 3 times competition technique were applied, totaling 15 knowledge models for each dataset. In this way, for every 3 models generated, the diagnosis was made and the metrics calculated, and at the end of the cross-validation process, an average for each metric was calculated.

The competition technique added about 5 to 15% to the final value of each metric. This is justified because the network creates independent models that learn in different ways, for the same dataset, in each execution. The order in which the input patterns feed the network and the value of its parameters influence how models learn. Thus, for each partition built by cross-validation, on each dataset, the algorithm is executed three times and, in the end, the result that occurs most frequently, for each input pattern, is the ANN prediction.

The Fuzzy-ARTMAP ANN used rapid learning in the training stage, using only one epoch for each knowledge model generated. This characteristic is one of the great advantages of the ART family algorithms concerning other approaches available in the literature, which generally use several epochs to converge.

Tests performed on the DB_prot dataset showed the highest performance ratings, all above 99%. However, the viral database (DB_vir), even with good rates, obtained the lowest values for all metrics, indicating that the Fuzzy-ARTMAP ANN had problems generalizing it. This indication can be seen in Table 5, where the number of neurons created for the virus taxon is higher than the others. Other corroborating evidence is the fact that viral epitopes have greater variability (Table 4) and a high mutation rate [43,44].

The same conditions used to create the training dataset were applied to the validation dataset, i.e., all proteins that tested positive in at least two assays were stored as epitopes, and all that tested non-positive in at least two assays were stored as non-epitopes.

Compared to the training data, the validation set has a higher proportion of bacteria verified regions (18.81% vs. 42.69% against 25.10%). Furthermore, the proportion of epitope versus non-epitope regions changed from 48.08% of epitopes in the training dataset to 34.62% in the validation dataset.

For the validation dataset, the knowledge models reached a sensitivity rate of 81.84%, even with the number of negative epitope residues much higher than the number of positive epitopes. Therefore, deterministically saying that all residues are negative epitopes may provide a high rate of specificity but not sensitivity. On the other hand, a specificity of 74.75% also ensures that the model did not predict all residues as positive epitopes, otherwise it would have very low specificity.

We evaluated the performance of BepFAMN with competing methods in the validation dataset for the verified regions, using the AUC of the ROC curve and the precision-recall curve. In both, BepFAMN outperformed competing tools’ prediction approaches. EpiDope, which had the best values for both metrics, was outperformed by 18% and 17%, respectively. The second best method (Beta-turn) was surpassed by 22.10% and 27%, respectively (Table 7).

These results indicate that BepFAMN was able to generalize the prediction task and can even predict data relatively distinct from the training data, with greater accuracy than competing methods.

## 6. Conclusions

This research has contributed by offering a new approach to solving the target problem based on ANN using adaptive resonance theory (ART). Although this architecture was proposed a long time ago (the 1980s), it is certainly current and quite competitive in relation to new techniques available in the specialized literature, such as “deep learning”. It should be noted that the ANNs of the ART family are also susceptible to the implementation of the “deep learning” concept and, in fact, it is an approach that is being developed by the research group.

In this sense, it opens the perspective of applications in several areas of human knowledge, including, in particular, continuing research in the health context, for example in aiding the development of immunizing agents. It is considered an important contribution to the technological area of public health, seeking to establish a healthy “symbiosis”, providing a great opportunity to develop new ANN architectures.

In this study, we developed a new method for predicting linear epitopes of B-cells, which, despite requiring only an amino acid sequence as input, has shown to have the best performance among the current tools used for this type of prediction.

The BepFAMN technique is based on an ANN from the ART family, and was trained on a dataset of approximately 21,000 experimentally verified epitope and non-epitope regions. We used the training dataset to understand the behavior of ANN on each taxon, and on the set of three taxon. In these sets, five-fold cross-validation was used to ensure the reliability of the experiment. Furthermore, all proteins in a subset have a sequence identity of less than 80% of the other proteins in the remaining 4 subsets.

The second dataset, the one used for validation, consisted of almost 25,000 new epitopes and non-verified epitopes, both of which were not present in the training dataset. We used this dataset to compare the performance of BepFAMN with other current tools available in the literature and frequently used [12,13,19]. In this set, BepFAMN outperformed all other competing methods (Table 7).

When analyzing the validation set, it is possible to notice that the specificity value was slightly below the sensitivity value, because the number of false negatives is greater than the number of false positives. This event demonstrates that ANN makes a more conservative forecast. In practice, a high true positive rate and a low false positive rate are much more important to reduce laboratory work.

The performance of BepFAMN can be attributed to the stability and plasticity characteristics provided by the ART family networks, its good generalization capacity for this type of data, and for the contribution of the competition technique. Furthermore, as the PSSM property was not used in the generation of the embeddings, we plan to develop a web and/or standalone version for the community, without a loss of performance.

Finally, epitope prediction tools should primarily serve as filters to rule out regions that are unlikely to be epitopes and, thus, eliminate unnecessary experimental analysis. In this way, new tools should emerge that increasingly improve sensitivity and specificity, in order to allow these experiments to be more precise and targeted.

## Figures and Tables

**Figure 1 sensors-22-04027-f001:**
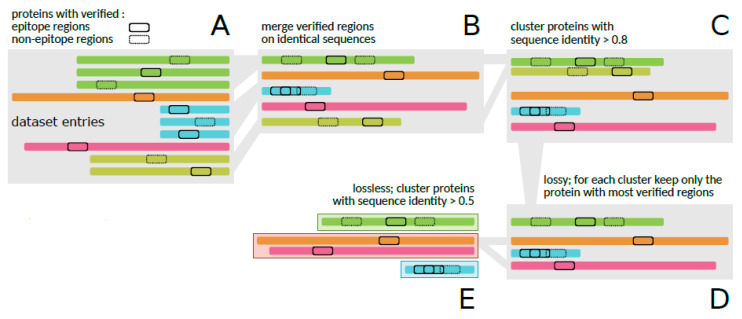
Pre-processing of data to generate the validation dataset. Initially, all amino acid sequences of identical proteins (**A**) are joined, preserving information about their verified regions (30,556) (**B**). In (**C**), all sequences that present a similarity greater than 80% are grouped in the same cluster. For each resulting cluster, only the sequence with the highest number of verified regions is selected (**D**). These selected sequences are then grouped again, but using a threshold of 50% identity (**E**) (adapted with permission from Ref. [13]).

**Figure 2 sensors-22-04027-f002:**
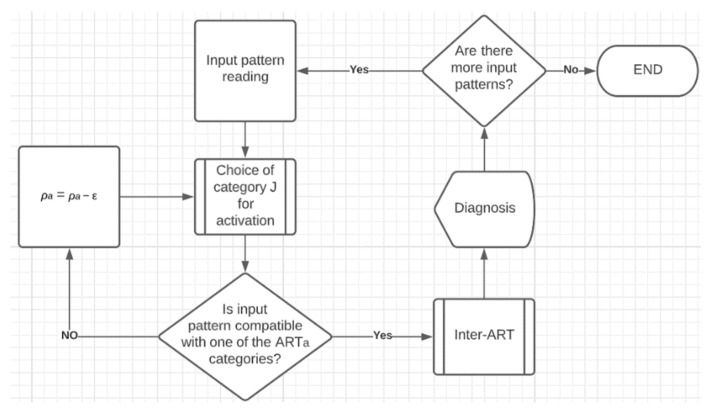
Flowchart of the FAM ANN diagnosis process.

**Figure 3 sensors-22-04027-f003:**
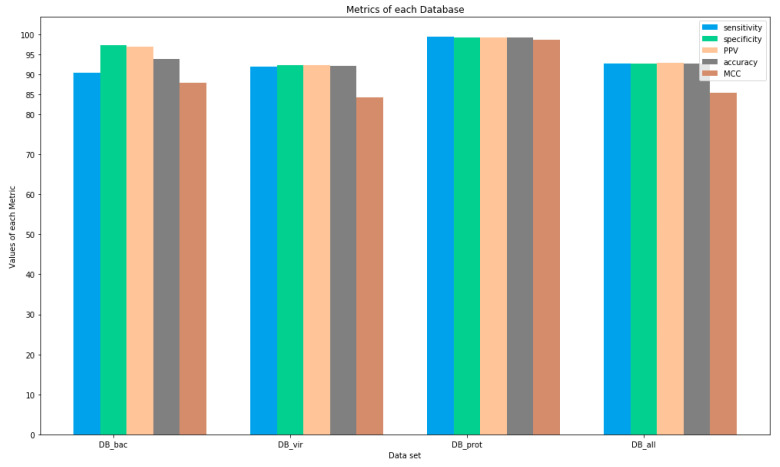
Overview of sensitivity, specificity, PPV, accuracy, and MCC metric results for each DB_bac, DB_vir, DB_prot, and DB_all test dataset.

**Figure 4 sensors-22-04027-f004:**
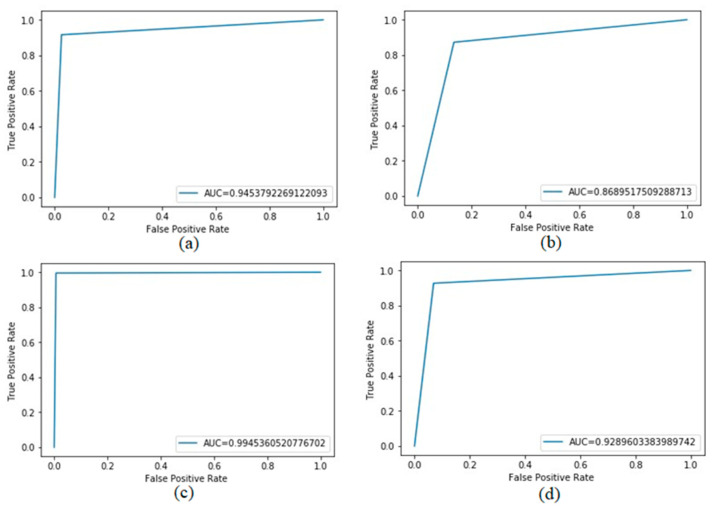
Area under the ROC curve using 5-fold cross validation for the following databases: DB_bac (**a**): database composed only of positive and negative bacterial epitope residues, DB_vir (**b**): database composed only of positive and negative virus epitope residues, DB_prot (**c**): database composed only of positive and negative epitope residues from protozoa and DB_all (**d**): database composed of positive and negative epitope residues from all three taxa.

**Figure 5 sensors-22-04027-f005:**
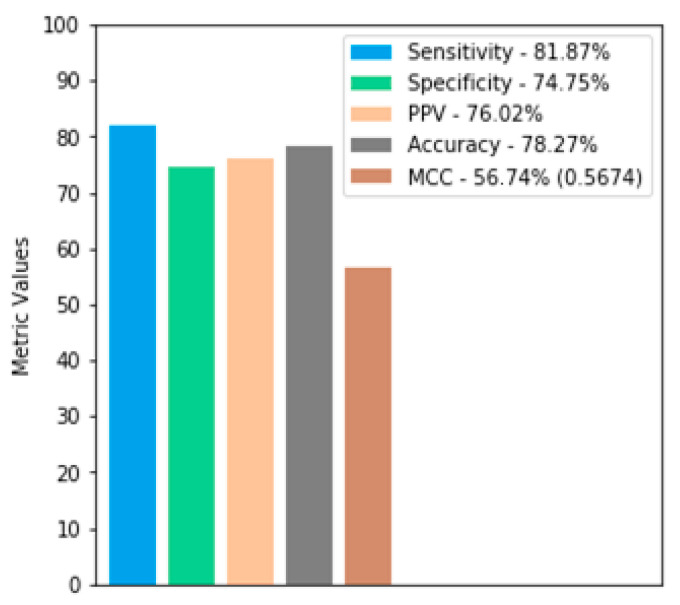
Results of the sensitivity, specificity, PPV, accuracy, and MCC metrics on the validation database using the DB_all model.

**Figure 6 sensors-22-04027-f006:**
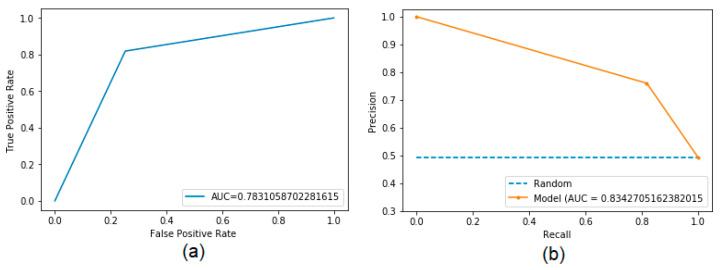
Area under the ROC (**a**) and precision-recall (**b**) curve using 5-fold cross-validation in the validation database with the DB_all model (database composed of positive and negative epitope residues from all three taxa).

**Table 1 sensors-22-04027-t001:** The number of positive/negative Epitopes extracted from the IEDB.

Taxon	Positive Epitopes	Negative Epitopes
Bacterium	1803 (15.67%)	4167 (14.84%)
Virus	5569 (48.39%)	6638 (23.64%)
Protozoan	4137 (35.94%)	17,275 (61.52%)
Total	11,509	28,080

**Table 2 sensors-22-04027-t002:** Original Database (adapted with permission from ref. [13]).

Original Database	
Verified Regions	30,556
Positive Epitopes	11,834
Negative Epitopes	18,722
Medium Length	15
Average of Length	13.50
Average Length of Positive Epitopes	13.99
Average Length of Negative Epitopes	13.20

**Table 3 sensors-22-04027-t003:** Number of positive/negative epitopes of each taxon after pre-processing.

Taxon	Positive Epitopes	Negatives Epitopes
Bacterium	1600 (16.05%)	2300 (21.36%)
Virus	4376 (43.90%)	4475 (41.57%)
Protozoan	3992 (40.05%)	3991 (37.07%)
Total	9968	10,766

**Table 4 sensors-22-04027-t004:** Mean and length variance of positive/negative epitopes.

Taxon	Positive Epitopes (Average)	Positive Epitopes (Standard Deviation)	Negative Epitopes (Average)	Negative Epitopes (Standard Deviation)
Bacterium	13.73	5.23	9.56	0.50
Virus	15.61	5.88	15.22	3.90
Protozoan	15.49	4.24	14.89	1.69
Total	15.25	5.22	13.88	4.03

**Table 5 sensors-22-04027-t005:** Neuron number of the DB_bac, DB_vir, DB_prot, and DB_all database knowledge.

Cross-Validation	Knowledge Model	Number of Neurons (DB_bac)	Number of Neurons (DB_vir)	Number of Neurons (DB_prot)	Number of Neurons (DB_all)
Partition 0	Model 0	210	1279	165	2256
	Model 1	216	1315	160	2250
	Model 2	209	1292	171	2270
Partition 1	Model 0	201	1340	193	2121
	Model 1	194	1265	205	2157
	Model 2	210	1321	186	2101
Partition 2	Model 0	203	1315	152	1845
	Model 1	189	1332	193	1816
	Model 2	216	1363	184	1801
Partition 3	Model 0	208	1237	170	1826
	Model 1	192	1258	171	1880
	Model 2	203	1295	191	1812
Partition 4	Model 0	198	1252	183	2044
	Model 1	188	1307	149	1998
	Model 2	199	1282	170	1943

**Table 6 sensors-22-04027-t006:** Metric results/dataset.

Dataset	Metrics				
	Sensitivity	Specificity	PPV	Accuracy	MCC
DB_bac	90.44	97.20	96.98	93.82	0.8788
DB_vir	83.42	87.39	86.88	85.40	0.7094
DB_prot	99.36	99.18	99.21	99.27	0.9855
DB_all	91.50	91.49	91.66	91.49	0.8300

**Table 7 sensors-22-04027-t007:** Comparison of the AUC of the ROC curve and of precision-recall curve between the tools (adapted with permission from Ref. [13]).

Tool	ROC AUC	Precision-Recall AUC
Proposed	0.783	0.83
EpiDope	0.605	0.66
BepiPred-2.0	0.465	0.56
Beta-turn	0.562	0.64
IUPred	0.550	0.60
Random	0.500	0.60

## Data Availability

The data used to support the findings of this study are available from the corresponding author upon request.

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
