# Peer review of "BepFAMN: A Method for Linear B-Cell Epitope Predictions Based on Fuzzy-ARTMAP Artificial Neural Network"

_sensors, 2022, doi:10.3390/s22114027_

Round 1

Reviewer 1 Report

According to the title and abstract of the paper by La Marca et al., this reviewer thought that the study would present a new way to better predict linear B-cell epitopes, and a new and improved B-cell epitope bank, which would be useful to researchers and biologists and other experts in human health. However, this very well written paper is very technical, and as a biologist, I am unable to evaluate the quality of the methods or the validity or originality of results.

Nonetheless, it is fine with me to publish this paper, to make the infomation available to those experts who are familiar with this type of studies.

Author Response

RESPONSE TO REVIEWER 1

According to the title and abstract of the paper by La Marca et al., this reviewer thought that the study would present a new way to better predict linear B-cell epitopes, and a new and improved B-cell epitope bank, which would be useful to researchers and biologists and other experts in human health. However, this very well-written paper is very technical, and as a biologist, I am unable to evaluate the quality of the methods or the validity or originality of the results.

Nonetheless, it is fine with me to publish this paper, to make information available to those experts who are familiar with this type of study.

We thank you for your consideration.

Reviewer 2 Report

This is a very interesting study and the engine seems to have a very high sensitivity, specificity, and accuracy. With the highest Precision-Recall, AUC compared to other popular servers. 

While Matthews correlation coefficient is a reliable evaluation matrix. A harmonic mean of precision and recall e.g. balanced F-score (F 1 score) could have served as a superior validation in this case where the training database and test databases have different categories and overlaps. 

This is a very impressive study. Can authors propose the application of this method in designing a similar server for discontinuous epitopes?

Authors have used training sets from experimentally validated epitopes/non-epitopes and are looking for agreement with their predictions vs. existing servers' predictions. For a more reliable testing training data set could have been randomly sampled from the experimentally established set and tested against the full set for a more stand-alone and novel matrix and not dependent on the performance of other servers. 

The computational material information is missing i.e machines, Operating systems, etc. Also, workflow managers, schematics, and program language information are also missing decreasing the value of this script as a method reference. 

Authors also need to disclose if this method will be made available to the public domain or if the modules are made available for evaluation by peers. 

Author Response

RESPONSE REVIEWER 2

Dear Reviewer, we greatly appreciate your relevant suggestions to improve this article. We to attend as best as possible, all the reviewers' recommendations, considering the deadline established by the Editor (5 days). Changes implemented in the article are highlighted in blue. However, we are available to attend if the answers were not satisfactory.

  1. This is a very interesting study and the engine seems to have a very high sensitivity, specificity, and accuracy. With the highest Precision-Recall, AUC compared to other popular servers.  While Matthews correlation coefficient is a reliable evaluation matrix. A harmonic mean of precisionand recall e.g. balanced F-score (F1 score) could have served as a superior validation in this case where the training database and test databases have different categories and overlaps. 

This suggestion was included in the article, that is, the values of F1 are presented.

  1. This is a very impressive study. Can authors propose the application of this method in designing a similar server for discontinuous epitopes?

The methodology proposed in this article is not applied to discontinuous epitopes. However, we may, on another occasion, develop a method applied, also, to discontinuous epitopes.

  1. Authors have used training sets from experimentally validated epitopes/non-epitopes and are looking for agreement with their predictions vs. existing servers' predictions. For a more reliable testing training data set could have been randomly sampled from the experimentally established set and tested against the full set for a more stand-alone and novel matrix and not dependent on the performance of other servers. 

Dear Reviewer: The experiment reported in the article was carried out as you indicated. One of the data sets was used for training and testing, while the other was used for validation.

  1. The computational material information is missing i.e., machines, Operating systems, etc. Also, workflow managers, schematics, and program language information are also missing decreasing the value of this script as a method reference. 

We understand your concern in this regard. However, the inclusion of suggested content is quite large, not including the space reserved for the preparation of this article.

  1. Authors also need to disclose if this method will be made available to the public domain or if the modules are made available for evaluation by peers.

This method will be available in the public domain.

Reviewer 3 Report

The purpose of the article “Title BepFAMN: A Tool for Linear B-Cell Epitope Predictions based on FUZZY-ARTMAP Artificial Neural Network” is to reported an the purpose of this work is to develop a tool that improves the accuracy of identifying linear B-cell epitopes. This research has contributed by offering a new approach to solving the target problem based on artificial neural networks using an adaptive resonance theory. The tool presented in this study can predict of B cells linear epitopes, which requires only the amino acid sequence as input. BepFAMN demonstrated the best performance among other frequently used tools for this type of prediction. The article, however, must be improved in terms of writing since some grammar and syntax errors are present in the manuscript. They should address the subject and critically review the information from the literature.

 My suggestions:

The authors need to revise the title of the paper in a more meaningful way.

Keywords are present in the title, choose others.

The abstract is written in a way lacks logic. It should highlight the salient findings more critically.

Introduction need more convincing rational for this article. 

The introduction has long paragraphs, I suggest reducing the size of the paragraphs.

The topic "2. Literature Review", I believe is unnecessary.

The results of this study are not fully explained therefore the interpretation of the results is very difficult. The author needs to provide the % increase or decrease rather than just writing ''significantly increased….''.

Table and Figures: Please provide standard error or standard deviation of the results, when necessary.

The discussion is poorly written hence, needs rewriting. The discussion should be further strengthened by adding some more relevant papers. The literature search is insufficient, only few related research papers in the past three years are cited, add the latest research results appropriately. See the below links if you think it will benefit your discussion.

Rewrite the conclusion! It needs to be much improved.

Author Response

RESPONSE TO REVIEWER 3

Dear Reviewer, we greatly appreciate your important suggestions to improve this article. Changes implemented in the article are highlighted in yellow.

  1. The purpose of the article “Title BepFAMN: A Tool for Linear B-Cell Epitope Predictions based on FUZZY-ARTMAP Artificial Neural Network” is to report the purpose of this work is to develop a tool that improves the accuracy of identifying linear B-cell epitopes. This research has contributed by offering a new approach to solving the target problem based on artificial neural networks using an adaptive resonance theory. The tool presented in this study can predict B cell’s linear epitopes, which requires only the amino acid sequence as input. BepFAMN demonstrated the best performance among other frequently used tools for this type of prediction. The article, however, must be improved in terms of writing since some grammar and syntax errors are present in the manuscript. They should address the subject and critically review the information from the literature.

       Suggested corrections have been included in the text. Also, a careful review of English has been provided.

My suggestions:

  1. The authors need to revise the title of the paper in a more meaningful way.

       The title has been corrected.

  1. Keywords are present in the title, choose others.

       This request has been included.

  1. The abstract is written in a way that lacks logic. It should highlight the salient findings more critically.

       The Abstract has been rewritten to meet this request.

  1. Introduction needs a more convincing rationale for this article. 

Also, the Introduction has been rewritten to meet this request.

  1. The introduction has long paragraphs, I suggest reducing the size of the paragraphs.

       Paragraphs have been revised.

  1. The topic "2. Literature Review", I believe is unnecessary.

  1. The “Literature Review” section has been deleted.

  1. The results of this study are not fully explained therefore the interpretation of the results is very difficult. The author needs to provide the % increase or decrease rather than just writing ''significantly increased….''.

       This request has been addressed in the text with due prominence.

  1. Table and Figures: Please provide standard error or standard deviation of the results, when necessary.

       This request has been granted.

  1. The discussion is poorly written hence, needs rewriting. The discussion should be further strengthened by adding some more relevant papers. The literature search is insufficient, only a few related research papers in the past three years are cited, add the latest research results appropriately. See the below links if you think it will benefit your discussion.

       This recommendation has been met in the text.

  1. Rewrite the conclusion! It needs to be much improved.

The “Conclusion” has been rewritten to meet this request.
